# Dealing with Alcohol-Related Posts on Social Media: Using a Mixed-Method Approach to Understand Young Peoples’ Problem Awareness and Evaluations of Intervention Ideas

**DOI:** 10.3390/ijerph20105820

**Published:** 2023-05-13

**Authors:** Hanneke Hendriks, Tu Thanh Le, Winifred A. Gebhardt, Bas van den Putte, Robyn Vanherle

**Affiliations:** 1Behavioral Science Institute (BSI), Communication & Media, Radboud University, 6525 GD Nijmegen, The Netherlands; 2Dutch Ministry of Health, Welfare and Sport, 2511 VX Den Haag, The Netherlands; 3Health, Medical and Neuropsychology Unit, Institute of Psychology, Leiden University, 2300 RB Leiden, The Netherlands; 4Amsterdam School of Communication Research (ASCoR), University of Amsterdam, 1018 WV Amsterdam, The Netherlands; 5Leuven School for Mass Communication Research, 3000 Leuven, Belgium

**Keywords:** alcohol posts, social media, intervention development, problem awareness, participatory action research

## Abstract

Young individuals frequently share and encounter alcohol-related content (i.e., alcohol posts) on social networking sites. The prevalence of these posts is problematic because both the sharing of and exposure to these posts can increase young individuals’ alcohol (mis)use. Consequently, it is essential to develop effective intervention strategies that hinder young individuals from sharing these posts. This study aimed to develop such intervention strategies by following four steps: (1) assessing young individuals’ problem awareness of alcohol posts, (2) unraveling individuals’ own intervention ideas to tackle the problem of alcohol posts, (3) examining their evaluations of theory/empirical-based intervention ideas, and (4) exploring individual differences in both problem awareness and intervention evaluations. To reach these aims, a mixed-method study (i.e., focus-group interviews and surveys) among Dutch high-school and college students (*N_total_* = 292, Age_range_ = 16–28 years) was conducted. According to the results, most youth did not consider alcohol posts to be a problem and were, therefore, in favor of using automated warning messages to raise awareness. However, these messages might not work for every individual, as group differences in problem awareness and intervention evaluations exist. Overall, this study puts forward potential intervention ideas to reduce alcohol posts in digital spheres and can therefore serve as a steppingstone to test the actual effects of the ideas.

## 1. Introduction

Research has shown that adolescents and young adults frequently use social networking sites (SNSs) to share updates about their leisure activities, including drinking activities [1,2]. Content analytic research, for example, shows that, on Facebook, 51% of young individuals from the Netherlands posted at least one picture showing alcohol (henceforth termed “alcohol posts”), and on Instagram, this was 28% [3]. The prevalence of these posts is worrisome because meta-analytic research shows that both the sharing of and exposure to alcohol posts across various SNS platforms (e.g., Facebook, Instagram, Snapchat, and WhatsApp) can increase individuals’ alcohol use [4]. Given the high percentage of alcohol use in general, and especially in the Netherlands, as well as the associated negative consequences (e.g., accidents, brain damage, and future alcohol addiction) [5,6], it is important to understand young individuals’ drive for and understanding of these posts to develop effective intervention strategies that can reduce the visibility and harmful effects of these posts. Although a few intervention approaches have focused on alcohol-related SNS use [7,8], these studies have mainly addressed how to use SNS to deliver intervention messages. Interventions aiming to reduce the visibility of alcohol posts are, to our knowledge, nonexistent. Consequently, this study aims to bring about potential intervention ideas by following four steps.

First, according to the literature, an important component of delivering an effective intervention is adequate problem awareness among the target group [9,10]. Although researchers in this domain agree that alcohol posts can have harmful effects [4,11], it remains unclear whether young individuals themselves are aware of the impact that alcohol posts may have on them. Consequently, little is known about whether young individuals actually perceive these posts to be problematic. Therefore, the first goal of this study was to gain more insight into the problem awareness of the target group by asking whether young individuals themselves are aware that alcohol posts and the posting of them involves potential harmful effects (RQ1). Second, building on the notions of participatory action research [12], another important component of developing effective intervention ideas is including the target group in all stages of intervention development. This was not frequently performed in previous research aiming to develop alcohol interventions, as young individuals are often only included in the testing phase [13,14]. Therefore, the second goal of the study consisted of actively involving young individuals from the beginning by addressing their own ideas for possible effective intervention ideas to reduce (the harmful effects of) alcohol posts on SNS (RQ2).

Third, apart from relying on participants’ perceptions, interventions are often built upon theoretical and empirical insights. However, although a lot of empirical evidence exists about the effects of alcohol posts [4,11], no research has, so far, used these insights to develop intervention strategies. In the third step, we created eight intervention ideas based on existing theoretical and empirical insights and evaluated the participants’ perceptions of these ideas (RQ3).

A fourth and final key factor to take into account when developing interventions is individual differences. Research indicates that interventions are not effective for everyone, as there is no one-size-fits-all intervention [7,15]. Consequently, in the final step, we aim to take these individual differences into account by examining how problem awareness and intervention evaluations might differ depending on personal characteristics (i.e., age, sex, education, and baseline drinking; RQ4). 

Altogether, these four goals were actualized by conducting a mixed-method study, consisting of qualitative focus-group interviews and a quantitative survey, among young Dutch individuals. By using such a mixed-method approach, we are able to shed light on the phenomenon from two different perspectives. On the one hand, the qualitative study allows us to capture in-depth individual perceptions, while the quantitative study, on the other hand, makes it possible to more systematically examine evaluations and individual differences. 

### 1.1. Effects of Alcohol Posts on Drinking Behavior

According to a previous meta-review [4], numerous studies have investigated the relationship between alcohol posts on SNS and offline drinking behavior among young individuals. This research often distinguished between self-sharing and exposure effects. 

First, research has shown that the sharing of alcohol posts on SNS can increase individuals’ drinking behaviors [11,16]. One explanation for these self-effects is suggested by the identity shift theory. According to this theory, individuals actively adapt their identities and behaviors based on how they present themselves in a given context, including the online context [17]. Consequently, after sharing alcohol posts, individuals might also drink more to align their established online drinking identity with their offline identity [11].

Second, apart from self-effects, research has also unraveled exposure effects by showing that exposure to alcohol posts affects youths’ drinking behavior [4]. Social norms are an important mechanism that explains this association [18]; that is, by seeing an alcohol post, people can get an idea of how many others are drinking (i.e., descriptive norms) [19], as well as how approving others are of this drinking (i.e., injunctive norms) [20,21], which, in turn, guides individuals’ behaviors. This aligns with the social learning theory, which shows that individuals imitate the behavior of significant others, even more so when the behaviors are performed by similar or desirable individuals and when they result in positive outcomes [22]. Thus, seeing an alcohol post showing peers who are having great fun together increases the likelihood of modeling such behavior [23]. 

The fact that the sharing of and exposure to alcohol posts have undesirable effects on youths’ drinking behavior makes these posts an urgent health concern. Thus, effective interventions that tackle the sharing of online alcohol posts are needed. This study addressed this need by following four steps.

### 1.2. The Importance of Problem Awareness

A first step in creating effective interventions, including alcohol-related interventions, involves increasing adequate problem awareness within the target group. For example, research on the promotion of sustainable travel behavior has shown that problem awareness is an important factor that influences people’s support for travel regulations [24]. In line with this, research in the context of alcohol has shown that awareness of one’s problematic alcohol-consumption patterns is an essential determinant in adjusting these patterns [25]. This can be explained by building on several socio-cognitive theories (e.g., Integrative Model of Behavior, Protection Motivation Theory, Health Belief Model, and EPPM), which argue that individuals’ perceptions of how likely they are to experience negative consequences of a behavior (e.g., consequences associated with problematic alcohol use) are essential in implementing recommended health behaviors (e.g., reducing alcohol use) [9,26,27]. These insights might also apply to alcohol posts, as it can be expected that young individuals who are more aware of the problem of alcohol posts and their negative consequences might also be more eager to adjust their posting behavior. Therefore, in the first research question, we aimed to gain more insight into young individuals’ problem awareness of alcohol posts by using both qualitative and quantitative data: 

**Research Question 1 (RQ1).** 
*To what extent are youth aware that alcohol posts can pose a problem (i.e., problem awareness)?*


### 1.3. Intervention Ideas to Decrease (the Effects of) Alcohol Posts

In a second step, we aimed to formulate a variety of potentially effective intervention ideas to decrease the posting of alcohol posts and limit their negative effects. To do this, a two-step approach was taken. First, based on insights from participatory action research [12], it is important to use a bottom-up approach and include youth in the entire research process, as this has proven to be beneficial for intervention outcomes. To illustrate, the results of a systematic review of youth substance-use prevention efforts showed that including youth in the research process increased community awareness of alcohol use and related solutions, thereby supporting the benefits of youth participation [28]. In a similar vein, a study on community-based health interventions showed that using participatory action research can be very valuable [29]. Moreover, in other contexts (e.g., LGBTQ+), participatory action research has been shown to be beneficial [30]. Building on these insights, the inclusion of youth might also be fruitful when developing interventions aimed at targeting alcohol posts. More importantly, young individuals are known as digital natives who might hold very different perceptions of SNS platforms and the content appearing on these platforms [31], hence showing the necessity to include them. In our research, we therefore actively involved the target group, that is, those who are affected by the problem, in the intervention development by letting young individuals suggest intervention ideas of their own to address the alcohol-post problem. This led to the second research question, which was answered by using qualitative data to ensure rich insights into young individuals’ perceptions:

**Research Question 2 (RQ2).** 
*Which intervention ideas are proposed by the youths themselves?*


In the third step, beyond the bottom-up approach, we used a top-down approach in which young individuals were presented with theoretical/empirical-based intervention ideas. The combination of both bottom-up and top-down approaches was chosen based on research advocating that both approaches can complement each other and, thus, provide richer insights [32]. In total, eight intervention ideas were developed. Ideas 1–5 were built on the mechanisms of self-sharing effects [17,33,34], in which we aimed to make the poster more aware of sharing alcohol posts and the possible consequences. Ideas 6–8 were built on the exposure effects of alcohol posts [4,11], in which we aimed to make the viewer more literate and capable of dealing with these posts. An overview of all ideas is provided in Table 1.

#### 1.3.1. Idea 1: Alcohol-Post Problem

As mentioned above, problem awareness is an important component of effective interventions. For example, awareness of one’s problematic alcohol use has been shown to be an essential determinant in adjusting one’s alcohol consumption [25]. However, when looking at alcohol posts, problem awareness seems to be lacking, as previous research showed that young individuals perceive alcohol posts as a positive and normal phenomenon [36]. Consequently, it is important to create awareness among young people that alcohol posts can, in fact, be problematic. One possible way of doing this is to provide statistical or factual information, as research shows that this can be fruitful in increasing knowledge and perceived vulnerability [14,37]. Thus, for the first intervention idea, we aimed to increase problem awareness by providing statistical information (e.g., research shows that seeing one alcohol post increases the odds of drinking alcohol by 15%). 

#### 1.3.2. Idea 2: (Too) Many Alcohol Posts

Aside from raising problem awareness, it is also important to make young people aware of their own posting behavior. In a study by Hendriks et al. [36], individuals mentioned that they often unconsciously share alcohol posts and are unaware of the fact that alcohol is present in a post. This was also shown in a recent study which revealed that young people have trouble remembering and accurately reporting their alcohol-posting behaviors and that, in general, young people spend “little thought” on the posting about alcohol [38]. Given these notions, it is important to make individuals realize that they are posting this type of content. This can be achieved, for example, by using self-monitoring techniques, as this appears helpful in changing health behaviors [39]. The second intervention idea, therefore, builds on self-monitoring by letting people count how many alcohol posts they have posted in the past.

#### 1.3.3. Idea 3: Warning Alcohol Posts

Another way to increase awareness of postings is the use of warning cues. Research in other SNS contexts has shown that warning labels can have beneficial effects. For example, Clayton et al. [40] showed that when labels warn readers that posts can contain misinformation, readers become more critical of the content they encounter. Warning labels seem especially effective when they stimulate thoughtful processing. For example, Kruijt et al. [41] showed that a simple message aimed to increase extensive processing (i.e., “Stop and think before trusting and sharing online information: evaluate the content and the source!”) helped people better recognize fake news. Regarding alcohol posts, this could be implemented with machine learning (that automatically recognizes alcohol in images) [42]. Thus, the third idea entails that when an individual is about to upload an alcohol post, he or she will receive a warning message, such as “You are about to upload a post that includes alcohol. Are you sure that you want to upload this?”

#### 1.3.4. Idea 4: Regret Alcohol Posts

In previous research, the posting of alcohol posts has also resulted in feelings of regret the day after, specifically when the post contained drunk and inappropriate content [43,44]. Anticipated regret has also been shown to increase binge-drinking intentions [45]. For the fourth intervention idea, we therefore focused on anticipatory regret by emphasizing that young people often regret “drunk sharing” the day after. 

#### 1.3.5. Idea 5: Perceived Identity of Alcohol Posts

Another important reason for regretting the sharing of alcohol posts is the fact that it can damage the self-presentation toward specific audiences [43,44]. Participants in a study by Vanherle et al. [1], for example, indicated deliberately limiting their alcohol posts for parents. Hendriks et al. [36] also showed that some young people do not post alcohol posts because they are afraid that future employers might see these. As such, for the fifth intervention idea, we aim to increase the importance of these audiences by emphasizing that alcohol posts are not always positively perceived by significant others, such as family members, parents, and future employers. 

#### 1.3.6. Idea 6: Correcting Misperceived Norms

As mentioned above, frequent exposure to alcohol posts has been proven to guide normative perceptions and more worrisome normative misperceptions [19,20]. This is problematic, as the overestimation of peers’ alcohol use might result in “drinking up” to perceived norms [8]. As a result, it is important to counter these normative misperceptions. According to Ridout and Campbell [8], Facebook private messages are useful in correcting normative misperceptions and reducing problematic drinking. Building on their insights, the sixth intervention idea consists of providing individuals with corrective normative messages.

#### 1.3.7. Idea 7: Alcohol Posts Are Unrealistic

Another possible idea is to counter the positive effects associated with alcohol posts. Research, for example, shows that alcohol posts reflect positive contexts and receive positive feedback [20,46], while negative alcohol-related effects (e.g., a hangover, fights, and harassment) are hardly shown [1]. This is worrisome, as the abundance of positive alcohol posts can communicate the idea that engaging in this behavior can result in positive outcomes for the viewer, thereby stimulating the modeling process [23]. Thus, building on the positivity-bias research [47], intervention Idea 7 aimed to make viewers more literate by making them realize that these posts do not showcase reality, as they are often too positive and do not show the negative sides of alcohol. 

#### 1.3.8. Idea 8: Popular Young People

As mentioned above, individuals may model the behavior displayed in alcohol posts and even more so when the model is a similar or high-status individual [22]. Important role models in this context may be social-media influencers (i.e., individuals with the potential to influence large audiences on SNS). Research shows that the use of social-media influencers is effective in boosting health campaigns [48]. Therefore, in intervention Idea 8, we made use of influencers to share negative information about alcohol on SNS. 

In summary, eight intervention ideas were developed based on previous empirical and theoretical insights. Before implementing specific approaches and testing the actual effectiveness of these ideas, it is important to assess how these ideas are perceived by the target group, as this benefits the actual effects of the interventions [49]. Thus, we asked a third research question, addressing it by using both qualitative and quantitative data:

**Research Question 3 (RQ3).** 
*How do youths perceive the theoretically proposed intervention ideas?*


### 1.4. Individual Differences in Problem Awareness and Intervention Evaluations

Although it is important to gain insight into young individuals’ evaluations of intervention ideas, it is also important to take individual differences in these evaluations into account. Research, for example, shows that the effects of interventions, as well as evaluations of perceived effectiveness, are not universal and can depend on personal characteristics [12]. Sex and age, in particular, have been found to influence the actual and perceived effectiveness of interventions [50,51]; however, education [52] and alcohol consumption [13] are also important factors. To illustrate, research has shown that individuals with high alcohol-consumption patterns, in comparison to individuals with low consumption patterns, experience different intervention effects [13]. Thus, it is important to study these individual factors, as it provides insight into what might (not) work for which specific individuals. Moreover, by unraveling important underlying factors, research could inform personalized intervention attempts, which have been proven to be a viable and cost-effective option for reducing problem drinking [7,15]. In particular, these personalized interventions might be especially valuable to tackle the problem of alcohol posts, as these are easily delivered online [13], the sphere in which these alcohol posts occur. Building on these insights, we therefore posed the fourth research question, which was addressed by using only quantitative data: 

**Research Question 4 (RQ4).** 
*What are the differences based on personal characteristics (in terms of age, sex, education, and alcohol use) in problem awareness and intervention evaluations?*


## 2. Methods

A triangulated mixed-method study consisting of qualitative semi-structured focus-group interviews and a quantitative survey was used to gain a thorough understanding of young individuals’ (problem) awareness of alcohol-related SNS postings and, relatedly, intervention methods to limit (the effects of) these postings. This mixed-method approach provides the possibility to study our research questions from different perspectives by simultaneously combining the rich subjective insights from our focus groups with the standardized, generalizable data from our surveys [53]. The two studies both heavily build on the notions of participatory action research [12], which highlights the importance of involving the target audience in different stages of intervention development, and which stresses that research should be performed “with” people, not “on” people. As argued by other researchers [29], it is very useful to employ mixed methods (using both quantitative and qualitative methods) within participatory action research, as this approach is more powerful to disentangle the complexities inherent to the research phenomenon [54]. We focused on Dutch adolescents (high-school students) and young adults (college students) in both studies, and ethical approval was received by the university’s ethical review board (2019-PC-11372).

### 2.1. Method Study 1 (Qualitative Study)

#### 2.1.1. Participants

In total, seven focus groups were conducted (*N* = 29). The high-school students were recruited with permission from a teacher in the northeastern region of the Netherlands, resulting in three focus groups at the same school, with six participants each (*n =* 18). A total of 2 participants were excluded from analysis, as they were younger than 16, resulting in a sample of 16 participants (10 women and 6 men, *M*_age_ = 16.75 years, range_age_ = 16–19 years). Of this sample, 10 students followed HAVO (general secondary education), and 6 followed VWO (pre-university education). 

For the university students, we relied on a network sampling approach in which participants were approached via Facebook or the personal networks of the second author. This resulted in four focus groups with two or three participants each (*n* = 11) (5 women and 6 men, *M*_age_ = 22.60 years, range_age_ = 19–28), all attending the same university. Five students were in a bachelor’s program, five were in a master’s program, and one was in a pre-master’s program. The detailed characteristics of the focus groups can be found in Appendix A. 

#### 2.1.2. Procedure

Participants had to provide active written consent before enrolling in the focus groups. The focus groups were conducted by the second author in November and December 2019. This author was female, 25 years of age, Dutch, and enrolled as a research master’s student at that time. The sessions took place in a classroom at the high school or university of the participants and lasted between 20 and 41 min. The focus groups were conducted in the native language (i.e., Dutch) of the participants and were audio-recorded with their written permission. 

A semi-structured interview guide was used for conducting the focus groups (Appendix B). First, participants had to report on their most frequently used SNS platform. Participants were then shown four alcohol-related posts (Appendix C) and were asked whether they posted or saw such posts from friends on SNS. Their perceptions and problem awareness of these posts were discussed by asking questions such as “What do you think when you see these alcohol posts?” and “To what extent do you think these posts are a problem?” Next, we became a bit more directive in the focus groups by providing participants with additional information (see Appendix B and Appendix C) to make them aware of the potential negative consequences of alcohol posts. Participants were then asked for ideas on how to limit the sharing and effects of alcohol posts. Finally, in a ranking exercise, participants were asked to rank the perceived effectiveness of the intervention ideas described above. See Table 1 for how we explained these intervention ideas to the participants. Participants started this exercise in dyads (or alone (in case of uneven groups)), followed by a group discussion in which they addressed what they liked and disliked about each idea. The whole group was then asked to jointly rank the three most and least effective ideas. After the discussion, participants were thanked, and six gift vouchers (EUR 10 each) were raffled among all participants.

#### 2.1.3. Analysis

The saturation point was used as an indication to stop data collection and was reached after seven focus groups. This is in line with the thematic analysis described by Guest et al. [39], who argued that three focus groups are generally considered sufficient to capture the most important themes. These focus groups were then transcribed (non-verbatim in Dutch) and anonymized by the second author, and then the last author started the coding process. Theoretical semantic thematic analysis [40] was chosen for this process because certain themes (prompted by the ranking exercise and evaluation of alcohol posts) were based on a priori constructed theoretical insights [41]. To be exact, the coding involved a social-constructivist approach, as the researcher actively steered participants’ discussions, as well as a naturalistic approach by simultaneously integrating open-ended questions that grasped participants’ internal reasoning and thoughts [42]. 

Overall, the following steps were taken: First, transcripts were divided into smaller fragments, using the data analysis software ATLAS.ti (version 8.4.22, 2019, open coding). These fragments were labeled with codes that described the participants’ thoughts. During the second stage (axial coding), possible links between the coded fragments were explored [40]. Related codes were then merged into overarching categories (e.g., Fear of Missing Out [FOMO]) and, eventually, in the final step, into themes. Three main themes were found based on participants’ reflections of the shown alcohol posts: prevalence, perceived appropriateness, and problem (see Coding Scheme 1 in Figure 1). In addition, the categorization of the intervention approaches resulted in five themes that aligned with the theoretical framework: problem awareness, regret, identity, media literacy, and socialization and modeling (see Coding Scheme 2 in Figure 2). Several techniques to enhance the validity of the results were used (see Appendix A).

### 2.2. Method Study 2 (Quantitative Study)

#### 2.2.1. Participants and Procedure

We distributed an online questionnaire by using the research panel *PanelClix* (for high-school students) and the research platform of the University of Amsterdam (for college students). After providing informed consent, participants could fill out the survey (which addressed concepts similar to Study 1, in the same order). After completion, the participants were debriefed and thanked for their participation. All participants provided written informed consent themselves because no participant was aged below 16 years (which would have required additional parental approval). The university’s ethics committee approved both Study 1 and Study 2 (ERP 2019-PC-11419).

In total, the combined sample consisted of 282 participants. Nineteen participants were excluded from the analysis because they completed less than 75% of the survey. This resulted in a final sample of 263 participants, consisting of 187 women (71.1%), 74 men (28.1%), and 2 non-binaries (*M*_age_ = 18.90 years, *SD*_age_ = 2.45, range_age_ = 16–28). Within this participant pool, 102 participants (38.8%) were minors (16–17 years), and 161 participants (61.2%) were 18+ years. In total, 51 participants (19.4%) studied at a high school, 23 participants (8.7%) followed vocational education, 18 participants (6.8%) studied at an applied university, 167 participants (63.5%) studied at a university, and 4 participants (1.5%) were currently not studying. Most participants consumed alcohol (i.e., 17.5% never consumed alcohol), and most participants (55.1%) had experience *sharing* alcohol posts themselves. Furthermore, most participants (91.6%) indicated that they had *seen* alcohol posts of friends regularly, with many of them indicated seeing these posts several times a week. 

#### 2.2.2. Measures

***Demographics***. We measured the participants’ sex (“What is your sex? *Male/Female/Other*”), age (“What is your age?”), and education (“What education are you following at the moment? *Practical Education/VMBO/HAVO/VWO/MBO/HBO/university*”).

***Alcohol Consumption*.** Participants’ alcohol consumption was measured by asking “How often do you normally consume alcohol?” on a 5-point scale (1 = *never*; 2 = *once a month or less*; 3 = *2–4 times a month*; 4 = *2–3 times a week*; 5 = *4 or more times a week*). 

***Problem Awareness of Alcohol Posts*.** Problem awareness was measured by asking the following: “To what extent do you think alcohol posts are a problem?” This question was answered on a 5-point scale (1 = *not a problem at all*; 5 = *very much a problem*). 

***Intervention Ideas Ranking and Motivation*.** Similar to Study 1, we first explained to participants that alcohol posts can pose a problem. We then asked participants to rank our eight developed intervention ideas from best (1) to worst (8). 

## 3. Results

### 3.1. Results Study 1 (Qualitative Study) 

#### 3.1.1. Alcohol Post Prevalence, Perceived Appropriateness, and Problem Awareness

Participants clearly distinguished between the four types of alcohol posts. Whereas post one (i.e., alcohol shown in the background, group of friends having a drink) and post two (i.e., focus on alcohol, glass of wine at dinner) were considered to be moderate and appropriate posts that showed a nice and sociable moment, post three (i.e., drunk post) and post four (i.e., drinking game post) were considered to be more excessive and oftentimes inappropriate posts because they could lead to possible reputation damage. 

Focus Group 5, university students

Ciara (Female, 25 years old):“Um, yeah. What do I think, I thought 1, that’s, I think it’s nice, just to see it. 2 also, specifically when someone has something to celebrate, for example, I think: ah, yes, nice! That’s just, it remains with one drink, so yes. But if you show a whole table, I don’t know, with all kinds of beer or alcohol, then I think, okay, what are you doing? And especially with a picture or a video of someone who is so wasted that he can’t even stand normally on his legs anymore. Then I really think, Yes, that is just super stupid.”

Participants mainly mentioned posting and seeing posts one and two in public social media environments (e.g., Instagram and Facebook). Posts three and four, on the other hand, appeared less frequently, and if they appeared, they were mostly posted in more private and temporary social media environments (e.g., WhatsApp and Snapchat). Given that moderate posts, which display alcohol subtly, were encountered most frequently, it was clear that most participants were not aware of the possible negative consequences of alcohol posts. 

Focus Group 4, university students

Atiyah (Female, 26 years old):“Yes. I think it’s tricky because young people probably don’t even know that they’re influenced by these posts. Like, I wasn’t even aware until you told me right now. Like, I wasn’t aware of that. Only when I started thinking about it, I was like: Oh yeah, wait, it could have an impact.”

However, some participants mentioned more negative sides of alcohol posts, as they indicated that the sharing of these posts was considered to be normal and often necessary to be part of the drunk fun. Consequently, some participants clearly experienced a fear of missing out after seeing these posts. Moreover, they said that these posts could further enhance drinking behavior because individuals would imitate the displayed behavior to adhere to the existing norm. 

Focus Group 1, high-school students

Andrea (Female, 18 years old):“The other day, I saw a photo of a birthday party where they had a lot of bottles of alcohol. And then you’re more inclined to think: ‘Oh, I’m also having a party soon, so I must have as much as they are having,’ because it’s a bit normal to have so much.”

#### 3.1.2. Intervention Ideas Proposed by Participants

Generating intervention ideas was considered a difficult task. Participants indicated that there was possibly no straightforward solution, mainly because alcohol posts were often very subtle and seemed to influence alcohol use indirectly. Still, participants raised a few new ideas: (1) making alcoholic drinks less aesthetically appealing so people would not take pictures of them, (2) implementing an age restriction algorithm so younger people will not see alcohol posts, (3) emphasizing that people who are enjoying the moment don’t have time to be on their phone, and (4) creating apps that could blur out the alcohol. 

Focus Group 5, university students

Ciara (Female, 25 years old):“I think you should just really start to actually reduce alcohol consumption in general, or maybe you should develop an app or something, so that you can, for example, start fading alcoholic drinks.”

Focus Group 1, high-school students

Andrea (Female, 18 years old):“Hey, but I wouldn’t take a picture of an ugly bottle either. You put that one in the back anyway.”

Interestingly, some participants explicitly stated that they would not implement intervention strategies, as this could result in a boomerang effect: by prohibiting someone from posting alcohol-related content, the poster might become even more interested in this content and subsequently post more. 

Focus Group 4, university students

Atiyah (Female, 26 years old):“Look, you have to give a convincing reason not to post something. I mean, you can say, ‘Yeah, it’s not cool!’ or whatever, but young people are always going to think booze is cool. Even more if you ban it. [Silence.]”

#### 3.1.3. Perceptions of Theoretically Proposed Intervention Ideas

In-depth evaluations of the intervention strategies that were discussed in the focus group are reported below in order of the eight intervention ideas (see Table 1). Idea 3 (warning alcohol post) was considered the most effective strategy, as this was the only strategy that appeared in the top three of all seven groups. By contrast, Ideas 1 (problem awareness), 4 (regret), and 6 (correcting normative misperceptions) were considered to be least effective.

##### Idea 1: Alcohol-Post Problem 

As mentioned above, most participants were unaware of the fact that alcohol posts could lead to negative effects. Therefore, the participants indicated that it would first be necessary to raise awareness. However, some participants agreed that providing statistics (e.g., “alcoholposts can increase the chance of drinking by 15%”) might not be the best way to raise awareness, because they can be perceived differently by individuals. For example, whereas some participants perceived the statistics (i.e., 15%) to be captivating, other participants did not perceive them as a threat at all. 

Focus Group 1, high-school students

Erin (Female, 17 years old):“I’m always sensitive to those percentages, though.”

Focus Group 3, high-school students

Jerry (Male, 17 years old):“I personally think that a 15 percent increased chance is not captivating for most people.”

Focus Group 6, university students

Hugh (Male, 20 years old):“15 percent might not say that much to people.”

Relatedly, participants mentioned that factual numbers lack a personalization aspect and, hence, might not capture the attention of every individual. In their opinion, it would be better to use visuals, humor, or a fun fact. Thus, the strategy might be more effective when conveying it in another way. 

Focus Group 3, high-school students

Tanya (Female, 16 years old):“Then yes, you should put a picture, but you shouldn’t put the sentence like ‘15 percent of young people have started drinking more,’ that would, ….just no. That doesn’t do anything for me.”

##### Idea 2: (Too) Many Alcohol Posts

Participants were generally not in favor of Idea 2 (counting alcohol posts), as this required too much effort from individuals. Moreover, they argued that just scrolling and counting the alcohol posts would probably not work because young individuals are often unaware of posting alcohol and, relatedly, the possible consequences of this posting. 

Focus Group 3, high-school students

Humberto (Male, 16 years old):“I don’t think it really does anything to anybody.”

Jerry (Male, 17 years old):“No, because when you see all those pictures, you don’t realize what the consequences were of those pictures.”

##### Idea 3: Warning Alcohol Posts

Most participants were in favor of using warning cues whenever someone was about to post alcohol content (Idea 3). This was considered to be an accessible strategy that would create awareness without being too didactic. The participants mainly liked the fact that it solely focused on you and your alcohol post. In addition, this strategy could inform the poster and protect the receiver at the same time.

Focus Group 1, high-school students

Chantal (Female, 17 years old):“I do think it’s a good idea. I also think that if, for example, every time you get a message saying, ‘You’re posting something with alcohol on it,’ then at a certain point, you realize, ‘OK, so there’s something wrong with it, and you’re already spreading information with it.’”

Andrea (Female, 18 years old):“So you inform the sender and the receiver of the post. That you slow it down a bit on both sides.”

Focus Group 6, university students

Hugh (Male, 20 years old):“I thought this machine learning, I thought that was pretty smart. I think, if you want to post something, and it says, ‘Yo. There’s also alcohol in here and stuff’ that you’d think about it for a second, one second longer.”

##### Idea 4: Regret Alcohol Posts

Participants were mainly negative about emphasizing feelings of regret after uploading alcohol posts (Idea 4), particularly because the post and the associated regret would only be temporary. Moreover, participants said they would more likely regret the negative consequences of drinking than the negative consequences of sharing the alcohol post: 

Focus Group 1, high-school students

Chantal (Female, 17 years old):“I think it’s a good idea, but I personally always have, the day after I’ve been drinking and I have a hungover, I always regret it. But it’s always a short moment; the next day it’s always over.”

Andrea (Female, 18 years old):“I always like it when I see those photos again. When I wake up in the morning and think, okay, that was a good party.”

Interviewer:“Then you don’t think, ‘Naaah, regret?’”

Andrea (Female, 18 years old):“Well, when everyone has seen that picture, then I always think ‘Okay.... this is great (sarcastic).’”

Chantal (Female, 17 years old):“I don’t know. But the regret passes so quickly, too.”

Erin (Female, 17 years old):“*Yeah*, *I have that too.”*

Barry (Male, 19 years old):“Yes, regret quickly turns into laughter. You start laughing about it, and then you take a picture like that again.”

##### Idea 5: Perceived Identity of Alcohol Posts

Idea 5, about emphasizing how one could be perceived by others after posting alcohol posts, was seen as useful. However, according to the participants, the effectiveness highly depends on the audience of the post. For example, on the one hand, participants had specific audiences they wanted to impress and, hence, kept in mind when posting about alcohol (e.g., future employers), but on the other hand, participants did not worry about the effects on some other audiences, such as friends, mainly because these close others often know you well and do not care about alcohol posts. If, however, the audience consisted of parents, the opinions were less unanimous. Whereas some participants paid a lot of attention to how they could be perceived by their parents, others did not care. 

Focus Group 5, university students

Ciara (Female, 25-years-old):“I also think that when you go out until very late every weekend and so on, your parents might already have an idea like, ‘Well, there’s no way she’s going to drink coke all night.’ Those people are already a little closer to you, so maybe you are less interested in what they think. Or then you don’t worry about it that much, unlike, for example, an employer of yours, or I don’t know, the in-laws or something. I have no idea, but yeah, people who are a little bit farther away from you.”

##### Idea 6: Correcting Misperceived Norms

Participants were less positive about Idea 6, which entailed correcting misperceived norms. Participants argued that this strategy could backfire because of individual differences. For example, the numbers might not be perceived as meaningful or a threat to some individuals, while giving non-drinkers the idea that everybody drinks. 

Focus Group 6, university students

Jan (Male, 22 years old):“Because I think a lot of young people have the opinion that if they go out and drink 0 to 4 drinks, that’s not an excessive amount at all.”

Focus Group 1, high-school students

Andrea (Female, 18 years old):“I think specifically this example doesn’t work because it says here that the majority of young people drink 0 to 4 drinks. And so it indicates that the majority of young people drink. Well, only 4, but they do drink.”

Furthermore, some individuals might already comply with this norm and still post alcohol posts. For example, one could have a glass of wine at dinner and post about this.

Focus Group 4, university students

Atiyah (Female, 26 years old):“You know, you can be a responsible drinker and still post alcoholposts. You don’t have to get wasted, because in your examples you had very decent ones (alcoholposts) too.”

##### Idea 7: Alcohol Posts Are Unrealistic

Participants seemed to be both positive and negative about Idea 7 (i.e., alcohol posts are unrealistic). They were positive because it tackled the positivity bias on SNS by reminding people that the negative sides of alcohol use also exist. Moreover, some participants proposed making this idea even more effective by making it personally relevant (e.g., including familiar people). 

Focus Group 3, high-school students

Tanya (Female, 16 years old):“*Well*, *it’s very confronting.”*

Lorenzo (Male, 16 years old):“*Yes*, *confronting.”*

Tanya (Female, 16 years old):“I think so. You first see a girl who has put on makeup, all pretty, she goes for a drink and the next thing she is throwing up.”

Interviewer:“This works for you?”

Tanya (Female, 16 years old):“Yes, yes, because I don’t think about it that often. When I see a picture, I think, ‘Oh! Fun moment!’ But then maybe half an hour later, she’s lying there arguing because they all had too much … Well, I don’t think about that.”

However, participants also stated that this strategy would probably not work because the depicted negative sides of alcohol were “unrealistic” and “unrepresentative,” as most participants mentioned that this would not occur to them. If it did occur to them, participants mentioned that it would be their own fault, and this, in turn, would make them more insensitive to the post. 

##### Idea 8: Popular Young People

Participants evaluated Idea 8 about the use of popular young people positively. This was mainly because popular people could function as role models. However, the participants mentioned that the strategy would only be effective with a credible role model as an endorser. If this were not the case, the message would be perceived as insincere and a possible advertisement. 

Focus Group 1, high-school students

Barry (Male, 19 years old):“This is already happening, right? That influencers are used? It’s good, though, because little kids look up to these kinds of influencers.”

Erin (Female, 17 years old):“Yes, but I don’t know if it helps?”

Barry (Male, 19 years old):“When you see Lil Kleine posting this.... Yeah, Lil Kleine is not credible, either.”

…

Gabrielle (Female, 17 years old):“I would not go for Lil Kleine because he is someone who often drinks himself anyway.”

Thus, participants were advised to opt for popular young people with a healthy lifestyle, such as “Doutzen Kroes” and athletes, rather than people associated with drinking. 

### 3.2. Results Study 2 (Quantitative Study)

#### 3.2.1. Problem Awareness Alcohol Posts

In line with Study 1, the majority stated that alcohol posts were not problematic. That is, 169 participants (64.26%) indicated that alcohol posts were “absolutely no problem” or “not really a problem.” Only 47 participants (17.87%) said that alcohol posts were “somewhat of a problem” or “very much a problem.” Forty-seven participants (17.87%) scored “neutral.” 

#### 3.2.2. Ranking and Perceived Effectiveness

To investigate how the participants evaluated the intervention ideas, we focused on the averaged ranking scores. Lower scores indicated better-ranked ideas (*1 = best-ranked idea to 8 = worst-ranked idea).* Similar to the focus groups, participants were most positive about the strategy that used automated warnings when people intended to upload an alcohol post (Idea 3; *Mdn* = 3, *M* = 3.41, *SD* = 2.18). Idea 1 (*emphasize alcohol posts as a problem*, *Mdn* = 4, *M* = 3.84, *SD* = 2.16) and Idea 5 (*emphasize how alcohol posts can be perceived*, *Mdn* = 4, *M* = 4.07, SD = 2.14) were also seen as relatively promising ideas. The worst idea was, according to the participants, Idea 4 (*regret alcohol posts*, *Mdn* = 6, *M* = 5.49, *SD* = 2.13; see also Table 2).

#### 3.2.3. How Problem Awareness and Ranking Depend on Characteristics

The relationships between age, alcohol consumption, and problem awareness and ranking were tested using correlation analyses, given the continuous nature of the variables. First, age was not related to problem awareness (*r* = −0.08, *p* = 0.201), but it was related to the ranking of Idea 3 (i.e., *warning*) (i.e., the older the participant, the lower the preference (i.e., higher ranking scores indicate lower preferences) for Idea 3, *r* = 0.24, *p <* 0.001). Correlations with the ranking of other ideas were not significant. Second, higher alcohol consumption was significantly correlated with lower problem awareness (*r* = −0.34, *p* < 0.001), but it was not significantly correlated to the ranking of interventions (all *r* < 0.09, all *p* < 0.135).

In addition, the relationships between sex, education, alcohol use, and both problem awareness and ranking were tested using (M)ANOVAs, given the categorical nature of some of the variables. Regarding sex (excluding the “other” category (*n* = 2)), there were no significant differences between men (*n* = 74) and women (*n* = 187) in terms of problem awareness (*F* = 0.41, *p* = 0.524). However, there was a significant difference between the ranking of Idea 4 (*regret; F* = 4.98, *p* = 0.026) and Idea 6 (*correcting misperceptions; F* = 4.82, *p* = 0.029) between women and men. That is, men (*M* = 5.03, *SD* = 2.46) ranked Idea 4 as a better idea than women (*M* = 5.67, *SD* = 1.96), whereas women (*M* = 4.83, *SD* = 2.13) ranked Idea 6 as a better idea than men (*M* = 5.47, *SD* = 2.14). Regarding education (excluding participants from “Practical education,” “VMBO,” and “other” due to low *n*)**,** there were significant differences between education levels in terms of problem awareness (*F* = 3.04, *p* = 0.018). Tukey’s post hoc tests revealed that university students (*n* = 167, *M* = 2.23, *p* = 0.009) and HBO students (*n* = 18, *M* = 2.17, *p* = 0.086) had significantly lower problem awareness in comparison with MBO students (*n* = 23, *M* = 3.00). Furthermore, there were (marginally) significant differences between the rankings of Ideas 3 (*F* = 3.36, *p* = 0.011), 5 (*F* = 2.07, *p* = 0.085), and 8 (*F* = 2.22, *p* = 0.067) between the different education levels. However, Tukey’s post hoc tests revealed no differences between specific education levels, except for a marginally significant difference (*p* = 0.074) in the ranking of Idea 8 (*popular people*) between secondary education (HAVO, *n* = 23, *M* = 5.74) and university students (*n* = 167, *M* = 4.40).

## 4. Discussion

The goal of the present study was threefold: to examine to what extent youths are aware that alcohol posts can pose a problem (RQ1); to examine youths’ own ideas for possible interventions to reduce (the effects of) alcohol posts, as well as their perceptions of theoretically proposed intervention ideas (RQ2 and RQ3); and to explore how problem awareness and intervention evaluations might depend on personal characteristics (RQ4). Three main findings can be distinguished. 

First, both the qualitative and quantitative studies showed that the majority of young people viewed alcohol posts as not problematic. This seems to be in line with the literature indicating that the posting of alcohol posts is a well-accepted social phenomenon and often happens without much conscious awareness [36]. This is worrisome because the literature shows that alcohol posts are in fact problematic, as both the sharing of and exposure to alcohol posts lead to increased drinking behaviors [4]. Consequently, a first future step may be to adequately raise problem awareness among young individuals (e.g., by increasing personal severity and susceptibility to the problem), because this is often a necessary component of effective interventions [9,27]. In our developed intervention ideas, we aimed to do this by using statistics; however, although this idea received overall positive evaluations, it was not considered effective by everyone in the focus groups, mainly because this approach lacked a personal aspect. It might therefore be interesting to investigate whether narratives, which often elicit emotion, are more effective in raising problem awareness [55]. 

The second finding is that, in both studies, young people considered automatic warnings to be the most effective strategy. Building on insights from the focus groups, participants especially liked this approach because it could be effective in creating awareness without being too didactic (i.e., others telling you what is right/wrong). When the participants were asked to come up with their own intervention ideas, they proposed a very similar idea to the automated warning: that alcohol could be automatically blurred in the posts. The fact that participants seem to favor ideas that emphasize external control is interesting, as it shows that young people like to have some help to stop their automatic tendencies to post about alcohol. Automated warnings are particularly helpful in this regard, as they trigger such a “stop and think response” and elicit more thoughtful processing [41]. Another advantage of this strategy is that it can decrease both sharing and exposure effects. That is, a warning can be triggered before someone posts an alcohol post, which consequently may decrease sharing behaviors, but a warning can also be applied to existing alcohol posts, thereby serving as a warning that could dampen exposure effects on viewers. 

However, before being able to implement such a strategy, research should further investigate whether such warnings can actually decrease the sharing of alcohol posts, as well as the undesirable exposure effects. Moreover, even if we can confirm that such warnings, in reality, yield desirable effects, they would still be several steps away from being implemented in practice. For example, to be implemented on the Facebook or Instagram API, we would need the company’s cooperation. Then again, the extent to which the platforms would be willing to cooperate with this is uncertain. There may be ways of circumventing this (e.g., by letting people install certain plugins in their browsers); however, these methods are cumbersome and present their own challenges. Thus, although the automatic warning idea seems fruitful, the actual implementation will be challenging. 

In contrast to automated warnings, which were deemed to be the most effective, the participants in both studies agreed that the idea of triggering anticipated regret was the least effective one. Contrary to theoretical evidence suggesting that posting alcohol posts induces feelings of regret [43,44], as well as the effectiveness of campaigns that trigger regret [56], participants in this study did not consider this strategy to be effective. These contradictory findings might be explained by the way young individuals create an online drinking identity. For example, whereas some research shows that young individuals share transgressive pictures as part of the drunk fun [2], other research shows that these drunk pictures can simultaneously damage self-presentation toward specific audiences, thereby resulting in regret [43]. Thus, whether an individual might experience regret likely depends on the underlying motives for sharing this particular content, which can differ for each individual. Inducing overall feelings of regret, regardless of the individual, might, thus, not be effective. This conclusion is in line with a study showing that although people generally do have regrettable alcohol-related experiences, these experiences do not necessarily lead people to consume less alcohol in the future. Thus, the authors conclude that focusing on regret may not be an effective intervention [57].

Building on these individual differences, our study makes a third and final contribution by showing differences in problem awareness and rankings based on the participants’ characteristics. Regarding problem awareness, we found that people who consumed more alcohol and students from university or HBO (versus lower levels of education) especially viewed alcohol posts as less problematic. This is in line with other studies showing that alcohol posts are especially a problem among university students [36]. Based on the differences in the ranking of ideas, we found that younger participants especially liked the automated warning idea and that women liked the strategy of anticipated regret more than men. This shows that it is important to take personal characteristics into account when developing interventions, as their effects are not necessarily one-size-fits-all. Thus, it could be worthwhile to target those with low problem awareness (e.g., university students). Alternatively, the intervention approach could be tailored to the target group (e.g., using automated warnings, especially for younger people). 

### Limitations and Future Research Suggestions

Although this study provides important insights for creating alcohol-post-related interventions, some limitations should be mentioned. First, the interviewer in the focus groups was approximately six-to-nine years older than the high-school students and of similar age to the university students. Consequently, this could have made high-school students more hesitant to express their opinions compared to college students. 

Second, qualitative research is often non-directional in nature, trying to explore a phenomenon without restricting potential answers from participants. In our study, we were a bit more directive. That is, we asked participants what they thought about alcohol posts; however, to address the problem awareness, we also asked them the extent to which they considered this to be a problem. Moreover, later on in the focus groups, we also explained to the participants that alcohol posts can have harmful effects. We needed to do this because we wanted the participants to be able to think about potential interventions. However, in doing so, we shifted the conversation in a specific direction. Future research could consider using a more open approach to studying this phenomenon. 

Third, based on participatory action research [12], an approach that has been shown to be effective in increasing healthy behaviors [28], we actively involved young individuals in suggesting and evaluating intervention ideas for the alcohol-post problem. Although we know from research that perceptions of intervention effectiveness are related to actual effects of interventions [49], we did not test the effects of our ideas on actual alcohol-post behaviors. The current study should, therefore, be seen as a steppingstone for future research aiming to implement effective interventions to deal with alcohol posts. 

Fourth, the results of this study might not be fully generalizable to other contexts for two reasons. First, the sampling procedure followed a non-probability sampling design; therefore, not all population members had an equal chance of participating in the study. Second, this study was conducted in the Netherlands. Although several of the findings are in line with the literature from other countries (e.g., Belgium and the United States [1,20]), the results might not fully hold in other cultural and geographical contexts in which other norms and rules about the use of alcohol exist. Thus, it is advised that the proposed intervention strategies of this study and young individuals’ perceptions thereof are also tested in other contexts to examine whether the results can be replicated or whether other intervention approaches are desired across contexts. 

## 5. Conclusions

This study aimed to investigate potential intervention strategies to address the problem of alcohol posts. In a mixed-method study (using focus groups and surveys), four questions were addressed, shedding light on young individuals’ (1) problem awareness of alcohol posts, (2) own intervention ideas, (3) evaluations of theory/empirical-based intervention ideas, and (4) individual differences. The main findings suggest that most youths did not consider alcohol posts to be a problem. As a consequence, this may have led participants to be in favor of using automated warning messages to raise awareness. In contrast, the intervention idea to use anticipated regret was considered to not be effective, possibly because the experience of regret can vary between individuals. Indeed, the results showed individual differences in problem awareness and intervention evaluations, indicating that it is important to take personal characteristics into consideration when developing interventions, as their effects are not necessarily one-size-fits-all. Through these results, this study generates potential intervention ideas to reduce alcohol posts on social media, which can serve as a steppingstone to test the actual effectiveness of the proposed interventions.

## Figures and Tables

**Figure 1 ijerph-20-05820-f001:**
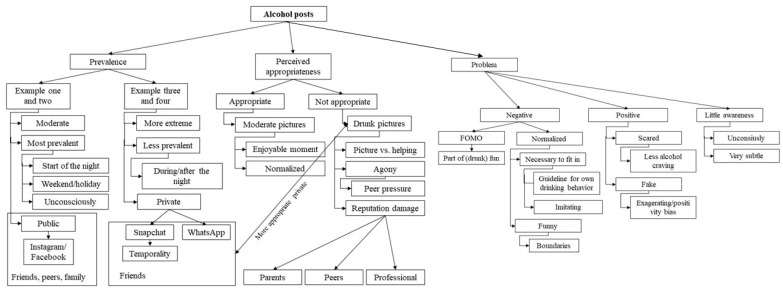
Perceptions and problem awareness of alcohol posts.

**Figure 2 ijerph-20-05820-f002:**
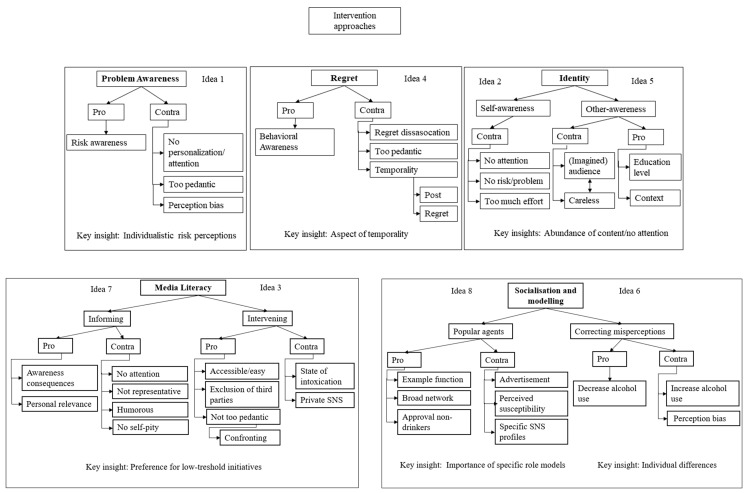
Perceptions of theoretical intervention approaches.

**Table 1 ijerph-20-05820-t001:** Eight intervention ideas and descriptions (as presented to participants).

Idea	Example Intervention Idea ^1^	Description
Idea 1: Alcohol-post problem	* 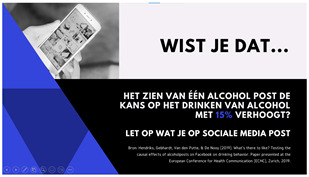 *	Create awareness among young people that alcohol posts can be problematic, for example, by providing facts/figures (“research shows that seeing one alcoholpost increases the odds of drinking alcohol by 15%”).
Idea 2: (Too) many alcohol posts	* 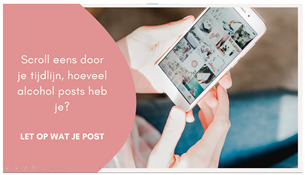 *	Allow young people to come to an understanding that they post (too) many alcohol posts, for instance, by letting them scroll through their timelines and count the number of alcohol posts.
Idea 3: Warning alcohol posts	* 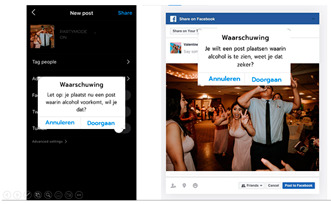 *	Provide an automatic warning when young people intend to upload an alcohol post (e.g., using machine learning to recognize alcohol in images automatically). For example, before uploading an alcohol post, they will get a message: “You are about to upload a post that includes alcohol. Are you sure that you want to upload this?”
Idea 4: Regret alcohol posts	* 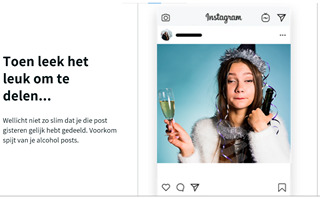 *	Emphasize that young people could regret sharing alcohol posts the day after uploading the posts.
Idea 5: Perceived identity of alcoholposts	* 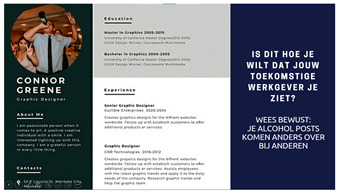 *	Point out to young people that alcohol posts are not well perceived (e.g., by parents and future employers). For example, show a resume including an alcohol post as a profile picture with the statement, “Is this how your future employer should perceive you?”
Idea 6: Correcting misperceived norms	* 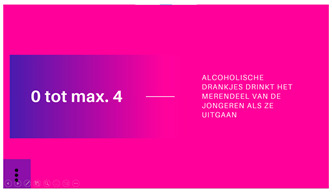 *	Young people unfairly hold the idea that many young people drink too much, which could be reinforced by alcohol posts. This idea could be corrected by messages on social media, such as “Most young people consume 0 to 4 drinks when they go out.”
Idea 7: Alcohol posts are unrealistic	* 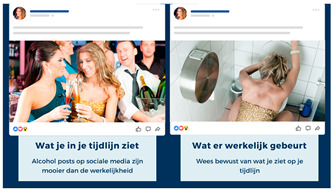 *	Show young people that alcohol posts are not an accurate representation of reality. These posts are often too positive and do not show the negative aspects of alcohol. For example, show a juxtaposition in which the first photo depicts “what you think happened” (laughing individuals with beers) and the second photo depicts “what actually happened after” (drunk individuals/vomiting).
Idea 8: Popular young people	^2^	Allow popular young people (e.g., good friends or influencers) to share social media messages that are negative about alcohol or emphasize not posting alcohol posts.

^1^ Two images of identical content were designed for each idea. When applicable, the second example portrayed individuals of the opposite sex. Below we provide the translations of the Dutch text in the images: Idea 1: Did you know that seeing one alcoholpost can increase the chance of drinking by 15%? Pay attention what you post on social media. Source: [35]. Idea 2: Take a scroll through your timeline: how many alcoholpost do you have? Pay attention to what you post. Idea 3: Warning: you are about to place a post containing alcohol, are you sure you want to do this? (cancel–continue). Idea 4: At the time it seemed fun to share. Maybe it was not so smart that you shared that post immediately yesterday. Prevent regret about your alcoholposts. Idea 5: Connor Greene (a fictional name with some fictional work-related information resembling a resume). Is this how you want your future employer to see you? Be aware of how alcoholposts are perceived by others. Idea 6: 0- Max 4 drinks are consumed by the majority of young people when they go out. Idea 7: What you see on your timeline. What really happened. Alcoholposts on your timeline are prettier than they really are. Be aware of what you see on your timeline. ^2^ The image of an influencer used as an example for this idea is available upon request (due to copyright reasons).

**Table 2 ijerph-20-05820-t002:** Ranking based on averaged ranked score, ordered from best- to worst-ranked idea.

	Ranking of Ideas		
	*M*	*SD*	*Mdn*
Idea 3 (warning alcohol posts)	3.41 ^a^	2.18	3
Idea 1 (alcohol-post problem)	3.84 ^b^	2.16	4
Idea 5 (perceived identity of alcohol posts)	4.07 ^b^	2.14	4
Idea 7 (alcohol post unrealistic)	4.39 ^b^	2.31	5
Idea 8 (popular young people)	4.65 ^b^	2.34	5
Idea 6 (correcting misperceptions)	5.02 ^c^	2.14	5
Idea 2 (too many alcohol posts)	5.12 ^c^	2.18	5
Idea 4 (regret alcohol posts)	5.49 ^d^	2.13	6

*Note.* Different superscript letters (e.g., a versus b) indicate (marginal) significant differences (based on paired *t*-tests).

## Data Availability

Data is unavailable do to privacy reasons.

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
