# Peer review of "Dealing with Alcohol-Related Posts on Social Media: Using a Mixed-Method Approach to Understand Young Peoples’ Problem Awareness and Evaluations of Intervention Ideas"

_ijerph, 2023, doi:10.3390/ijerph20105820_

Round 1

Reviewer 1 Report

Thank you for the invitation to review this paper. Please, find bellow my comments and recommendations:

1. I would recommend to the authors to write the abstract of the paper in order to make it more appealing for the readers.

2. The first section should be named ``Introduction`` and also should have a number. In this first part, I consider that the authors should better present the research context and also the research gap. I would suggest that the authors present some statistics regarding the issue.  Also, I consider that this phenomenon has a different impact on different cultures and countries. Therefore, I would advice the authors to clearly present the research context and significantly improve this part of the paper.

3. The literature review should be developed by consulting more relevant papers regarding each RQ. The authors should riguros use the citing style of IJERPH.

4. From a methodological perspective, the paper rises some questions that need to be addressed:

4.1 Why was this mixed research design necessary? As a reader, I had the perception that the authors didn't argued their methodological decisions.

4.2 The aim of qualitative research is to explore the phenomenon - therefore, some questions from the FG were actually trying to measure some variable (e.g., ``to what extent``). I consider that the author should be careful when using this type of questions and analyzing the data.

4.3 The survey was conducted using a nonprobabilistic sample - thererefore, generalizations can not be made based on these results. I would recommend that the authors be cautious when analyzing data and presenting the results.

4.4 Some analysis should be better argued: the correlations were possible between the variables considering the nature of the scales (i.e., parametric and nonparametric)?

The group analysis should be detailed - more information in necessary about the used sampling.

5. The conclusion of the paper should be more clearly presented. Also, the limitation of the study should be expanded based on my previous comments.

Good luck!

Author Response

Please see attachment, focusing on the part "Reply to comments of reviewer 1"

Reviewer 2 Report

First of all, I would like to congratulate the authors for the approach taken in this study.

There are few investigations that address alcohol consumption from a perspective mixed, and much less in adolescents.

Through this study, the important problem that is the consumption of alcohol in adolescents and how it is completely normalized within this group.
After reading the manuscript, I would like to clarify a series of aspects that I consider positive for research.

The intro section reflects alcohol posts on social media. Que profiles were analyzed?

In the procedure section, the request for consent of the participants Was this consent oral or written? There was a different consent for those minor participants? Were the parents or legal guardians of these children considered? participants, at the time of participating in the research?

Finally, I consider it relevant given the context of the investigation, in its result it is reflects how adolescents do not consider consumer publications as a problem, however recent research has shown that the publications of Moderate consumption are posted more frequently than extreme posts. Has this fact also been visible in your research?

Without further ado, thank you for the work done.

Author Response

Please see attachment, focusing on the part "Reply to comments of reviewer 2"

Round 2

Reviewer 1 Report

Thank you for sending me the revised version of the paper. The authors made significant changes to the initial draft of the paper, but, from my point of view much work still must be done:
- to consult more relevant references - the paper still needs a more solid theoretical support
- the research design should be better argued 
- the techniques researchers used to analyze the data could be inserted in the main body of the text (there are important aspects that should be moved from the Appendix) 

- the conclusion section should be expanded

Good luck!
